# Unraveling the lncRNA-miRNA-mRNA Regulatory Network Involved in Poplar Coma Development through High-Throughput Sequencing

**DOI:** 10.3390/ijms25137403

**Published:** 2024-07-05

**Authors:** Zihe Song, Chenghao Zhang, Guotao Song, Hang Wei, Wenlin Xu, Huixin Pan, Changjun Ding, Meng Xu, Yan Zhen

**Affiliations:** 1State Key Laboratory of Tree Genetics and Breeding, Co-Innovation Center for Sustainable Forestry in Southern China, Nanjing Forestry University, Nanjing 210037, China; zihesong_njfu@outlook.com (Z.S.); zhangchenghao@njfu.edu.cn (C.Z.); sgtendeavor@163.com (G.S.); wenlinxu.njfu@outlook.com (W.X.); njfuhxpan@163.com (H.P.); xum@njfu.edu.cn (M.X.); 2State Key Laboratory of Tree Genetics and Breeding, Key Laboratory of Tree Breeding and Cultivation of State Forestry Administration, Research Institute of Forestry, Chinese Academy of Forestry, Beijing 100091, China; wh18187082671@163.com (H.W.); changjunding@caf.ac.cn (C.D.)

**Keywords:** strand-specific RNA-seq, long non-coding RNA, miRNA, poplar coma, network

## Abstract

Poplar coma, the fluff-like appendages of seeds originating from the differentiated surface cells of the placenta and funicle, aids in the long-distance dispersal of seeds in the spring. However, it also poses hazards to human safety and causes pollution in the surrounding environment. Unraveling the regulatory mechanisms governing the initiation and development of coma is essential for addressing this issue comprehensively. In this study, strand-specific RNA-seq was conducted at three distinct stages of coma development, revealing 1888 lncRNAs and 52,810 mRNAs. The expression profiles of lncRNAs and mRNAs during coma development were analyzed. Subsequently, potential target genes of lncRNAs were predicted through co-localization and co-expression analyses. Integrating various types of sequencing data, lncRNA-miRNA-TF regulatory networks related to the initiation of coma were constructed. Utilizing identified differentially expressed genes encoding kinesin and actin, lncRNA-miRNA-mRNA regulatory networks associated with the construction and arrangement of the coma cytoskeleton were established. Additionally, relying on differentially expressed genes encoding cellulose synthase, sucrose synthase, and expansin, lncRNA-miRNA-mRNA regulatory networks related to coma cell wall synthesis and remodeling were developed. This study not only enhances the comprehension of lncRNA but also provides novel insights into the molecular mechanisms governing the initiation and development of poplar coma.

## 1. Introduction

Poplars, crucial fast-growing industrial timber and afforestation trees in the Northern Hemisphere [1], are widely used in papermaking, biofuels, pharmaceuticals, and construction [2]. China boasts the world’s largest artificial poplar forest, which has brought substantial benefits but also significant environmental and safety issues. In spring, female plants undergo double fertilization, leading to the development of capsules. Upon reaching maturity, these capsules discharge seeds accompanied by their fluff-like appendages, scientifically referred to as ‘coma’, which originate from the differentiated surface cells of the placenta and funicle [3]. Although the coma plays an indispensable role in the long-distance dispersal of seeds, it also has the potential to serve as a medium for the transmission of pathogens and pests. On the one hand, contact with these comas may result in allergic reactions and respiratory illnesses for some individuals, posing risks to the health of surrounding vegetation. Additionally, the flammable nature of comas increases the risk of fire occurrences. On the other hand, the substantial dispersion of comas also contributes to a certain degree of soil and water pollution in the vicinity, as well as impacting the urban and rural environment. To address this issue fundamentally, it is crucial to investigate the molecular mechanisms underlying the development of poplar coma. Both Arabidopsis trichomes and cotton fibers have been extensively studied regarding the mechanisms that regulate cell differentiation and polar growth. This has provided valuable references for our research.

Arabidopsis single-celled trichomes provide an excellent model for studying cell fate determination. Over the past two decades, the relevant molecular mechanisms have been elucidated. The initiation of trichomes is cooperatively regulated by multiple transcription factors, with the MYB-bHLH-WDR (MBW) transcription activation complex at its core [4]. The MBW complex can, on the one hand, initiate trichome formation by activating the expression of downstream *GLABRA2* (*GL2*) and *TRANSPARENT TESTA GLABRA2* (*TTG2*). On the other hand, the MBW complex can also trigger the expression of members belonging to the R3 MYB family. R3 MYB moves to neighboring cells, competing with R2R3 MYB for bHLH binding sites, hindering the formation of the transcription activation complex and suppressing trichome initiation [5]. In Arabidopsis, the development of leaf trichomes involves several stages, including the cessation of cell division, stemward expansion, outward protrusion, elongation, and branch formation, culminating in mature trichomes [6]. The cytoskeleton organization serves as the foundation for the morphogenesis of Arabidopsis trichomes, and disrupting the cytoskeleton can result in varying degrees of defects in trichome development [7,8,9]. Cotton fibers, distinct from the inception of poplar coma, derive from trichome primordia on the ovule epidermis. Their development encompasses four stages: differentiation, elongation, secondary cell wall formation, and maturation [10]. Cotton fiber initiation shares a cell fate determination model similar to the trichome formation in Arabidopsis [11]. As an exemplary single-cell model for studying cell elongation and cellulose synthesis, the functions of various genes regulating fiber elongation in cotton have been elucidated, including cytoskeleton-related genes (e.g., *GhTUB13* [12], *GbAct1* [13], and *GhKCH1* [14]); cell wall material synthesis-related genes (e.g., *GhCesA4*, *GhCesA7*, *GhCesA8* [15], *GhSUS2* [16], and *GhGLU18* [17]); expansin (*GbEXPATR* [18]), involved in cell wall remodeling; calmodulin (*GhCaM7-like* [19]) for Ca^2+^ signal transduction; ascorbate peroxidase (*GhAPX1* [20]) for ROS homeostasis; and lipid transfer proteins (*GhLTP4* [21]) for cuticle synthesis. In addition, plant hormones also play a crucial role in the elongation of fiber cells, with gibberellins (GA) [22], jasmonic acid (JA) [23], auxin (IAA) [24], ethylene (ETH) [25], and brassinosteroids (BR) promoting fiber development [26], while cytokinins (CK) [27] and abscisic acid (ABA) [28] inhibit fiber growth.

An increasing number of protein-coding genes have been demonstrated to be involved in the initiation and development of Arabidopsis trichomes and cotton fibers, but there is limited research on lncRNAs in this context. Long non-coding RNA (lncRNA) is a class of non-coding RNA exceeding 200 nucleotides in length. Typically characterized by low expression levels, it frequently exhibits tissue and cell-type specificity [29]. For an extended period, lncRNA was regarded as transcriptional “noise”; however, recent research in the past few years has revealed that lncRNA plays a role in plant growth, development, and stress responses through diverse mechanisms [30]. lncRNA often exerts its functions by regulating gene expression, encompassing three modes: pre-transcriptional, transcriptional, and post-transcriptional. Before transcription, lncRNAs can mediate gene expression by altering three-dimensional chromatin conformation. For instance, lncRNA *AUXIN-REGULATED PROMOTER LOOP* (*APOLO*) regulates *ROOT HAIR DEFECTIVE 6* (*RHD6*) [31]. They can also recruit histone modification factors before transcription, as exemplified by *LAIR* controlling *LRK1* [32], or bind to relevant gene promoters, such as lncRNA *Hidden Treasure 1* (*HID1*) regulating *PHYTOCHROME INTERACTING FACTOR 3* (*AtPIF3*) [33]. During transcription, lncRNAs regulate gene expression through transcriptional interference. For instance, lncRNA *SVALKA* influences the expression of *C-repeat/dehydration-responsive element binding factors (CBF)* [34]. Post-transcriptionally, lncRNAs can competitively bind to miRNAs to regulate gene expression. For example, lncRNA *LNC1* controls *SQUAMOSA PROMOTER BINDING PROTEIN-LIKE 9* (*SPL9*) expression through competitively binding to miR156a [35]. Furthermore, lncRNAs can interact with proteins, as demonstrated by *MISSEN*, which negatively regulates rice endosperm development by competitively binding to helicase family protein (HeFP) and inhibiting its interaction with tubulin [36]. lncRNAs can also serve as scaffolds, bringing together various proteins to regulate normal biological processes, including ribosome biogenesis [37].

Although wood characteristics and stress resistance in poplar have been extensively studied in breeding research [38,39,40], there is limited research on coma. This study employed strand-specific RNA-seq to identify lncRNAs and mRNAs across the entire genome during poplar coma development. Potential target genes of lncRNAs were predicted through co-location and co-expression analyses. Different types of sequencing data were integrated to unveil lncRNA-miRNA-mRNA regulatory networks in the development of poplar coma. Viewing poplar coma development through the lens of lncRNAs provides a scientific foundation for understanding the associated molecular regulatory mechanisms.

## 2. Results

### 2.1. Identification and Characterization of lncRNA and mRNA during the Development of Poplar Coma

To identify lncRNAs and mRNAs associated with coma development and achieve a more profound comprehension of the molecular regulatory mechanisms governing coma development in poplar, samples were gathered from three distinct developmental stages of poplar coma (W denotes the initiation stage of coma, P represents the early elongation stage, and Y represents the late elongation stage) to construct strand-specific libraries. High-throughput sequencing was then conducted. Nine libraries amassed 798,569,106 raw reads and 774,841,556 clean reads in total (Appendix A). A total of 79.80% of clean reads from the nine libraries were successfully aligned to the *P. trichocarpa* genome. Among these, 75.9% of reads were precisely mapped to a single location in the genome, while 3.9% were aligned to multiple positions. Specifically, 80.61% of reads were mapped to exonic regions across all samples, 15.82% were aligned to intronic regions, and 3.57% were aligned to intergenic regions. The results of the statistical analysis further indicate the high quality of the RNA-seq data.

Using the StringTie software, a total of 113,171 transcripts were assembled for subsequent identification of mRNA and lncRNA. Initially, comparing and annotating the assembled transcripts against known databases resulted in the identification of 52,810 mRNA transcripts. Following this, 59,051 transcripts were obtained by filtering those with a length equal to or exceeding 200 nucleotides from the remaining transcripts. CPC, CNCI, PFAM, and PLEK were employed to further predict the protein-coding potential of transcripts. The intersection of predictions from these tools yielded 12,832 transcripts. To refine the analysis, filtering was applied based on expression levels, selecting transcripts with FPKM values ≥ 0.5 for multi-exon and ≥2 for single-exon transcripts. This process led to the identification of 1888 lncRNAs (Appendix A). Categorizing these lncRNAs based on their positional information in the genome revealed four distinct types. Antisense lncRNAs numbered 474 (49.53%), lincRNAs numbered 7020 (39.79%), sense overlapping lncRNAs numbered 1633 (9.26%), and intronic lncRNAs numbered 250 (1.42%). Analysis of different types of lncRNA on each chromosome revealed that antisense lncRNAs, lincRNAs, and overlapping lncRNAs are universally generated on all chromosomes. However, chromosomes 5, 7, 8, 9, 10, 12, 14, and 17 do not produce intronic lncRNAs. Additionally, most chromosomes, such as 1, 3, and 4, exhibit a higher abundance of lncRNAs in regions closer to the ends, while chromosomes 2 and 7, among others, tend to generate more lncRNAs in the central regions (Figure 1). Examining their expression levels during the three developmental stages of coma indicated that intronic lncRNAs and lincRNAs exhibit relatively higher expression levels among the four different types of lncRNAs. Following them are sense-overlapping lncRNAs, with antisense lncRNAs exhibiting the lowest expression levels (Figure 2A). Further analysis of their characteristics uncovered that intronic lncRNAs exhibit an average length shorter than the other three types of lncRNAs, boasting the fewest exons and the shortest open reading frame (ORF) length. Sense-overlapping lncRNAs, on the other hand, showcase the longest average length, while antisense lncRNAs feature the highest average number of exons and, correspondingly, the longest average ORF length (Figure 2B–D).

Compared with mRNA, lncRNA demonstrates lower expression levels (Appendix A). The average mRNA length is 1443 bp, while the average length of lncRNA is only 698 bp, with 83.6% of lncRNAs having a length of less than 1000 bp. In contrast, mRNAs with a length below 1000 bp make up only 40.6% (Appendix A). The average number of exons for lncRNA is 1.9, whereas for mRNA, it is 5.9. Among lncRNAs, 79.8% have one or two exons, while only 33.9% of mRNAs have one or two exons (Appendix A). Additionally, comparing the ORF lengths of both, it was found that the average length of lncRNA ORFs is approximately 91 bp, while the average length of mRNA ORFs is around 478 bp. Only 6.2% of mRNA ORFs have a length of less than 100 amino acids (aa), whereas 73.2% of lncRNA ORFs have a length of less than 100 aa (Appendix A).

### 2.2. The Transcriptional Expression Profile during Coma Development

The FPKM method was employed to compute the expression levels of lncRNA and mRNA in each sample, conducting pairwise comparisons across the three stages of coma development. The results reveal that among the DELs in the PvsW combination, 12 were upregulated and 8 were downregulated. In the DEGs, 167 were upregulated, and 14 were downregulated (Figure 3A). Among the DELs in the YvsP combination, 47 were upregulated and 28 were downregulated, whereas in the DEGs, 2985 were upregulated and 908 were downregulated (Figure 3B). In the YvsW combination, there were 312 upregulated lncRNAs and 205 downregulated lncRNAs, with 9360 upregulated mRNAs and 6872 downregulated mRNAs (Figure 3C). In various comparison groups, the count of upregulated lncRNAs and mRNAs surpasses that of the downregulated counterparts. Furthermore, there are 5 lncRNAs and 762 mRNAs specifically expressed during the W period, 2 lncRNAs and 394 mRNAs specifically expressed during the P period, and 12 lncRNAs and 387 mRNAs specifically expressed during the Y period. To confirm the reliability of DELs, nine DELs were randomly selected and examined by RT-qPCR, and results were consistent with RNA-seq data (r > 0.88, Figure 4).

Delving deeper into the expression patterns of lncRNA and mRNA at various stages of coma development, a time-series analysis was performed on DELs and DEGs. The analysis of lncRNA unveiled that cluster 4 and cluster 5 exhibited the highest enrichment in terms of lncRNA. The expression of these lncRNAs demonstrated an upward trend during the progression of coma (Figure 3D). In the analysis of mRNA, clusters 3 and 7 were found to be the most enriched in terms of mRNA. The expression of these mRNAs showed a decreasing trend during the development of the coma (Figure 3E). Overall, there were distinct differences in the expression patterns between DELs and DEGs.

### 2.3. The Prediction and Functional Analysis of lncRNA Target Genes

The action of lncRNA on target genes can be categorized into cis- and trans-regulation. Trans-regulation, also known as co-expression, refers to the manner in which lncRNA, after transcription, regulates target genes located at a distance. To ascertain the potential target genes of lncRNA trans-regulation, an analysis of the expression correlation between different samples was conducted. The results indicated that 1310 lncRNAs were co-expressed with 15,754 genes (Appendix A). Conducting GO analysis on the target genes of DELs in each comparison group, significant enrichment terms were observed in the P vs. W group, including monooxygenase activity (GO:0004497), protein dimerization activity (GO:0046983), and oxidoreductase activity (GO:0016705), among others (Appendix A). In the Y vs. P group, enriched terms comprised catechol oxidase activity (GO:0004097), ribosome (GO:0005840), and pigment biosynthetic process (GO:0046148), among others (Appendix A). Finally, the Y vs. W group showed enrichment in terms such as kinesin complex (GO:0005871), microtubule motor activity (GO:0003777), and microtubule-based movement (GO:0007018), among others (Appendix A).

Cis-regulation, also known as co-location, refers to the potential regulatory effect of lncRNA on proximal protein-coding genes. Identified within a 100 kb range upstream and downstream of lncRNA were potential target genes for cis-regulation. The results showed that within a 10 kb range, 1808 lncRNAs targeted 4137 genes, within a 10–50 kb range, 1872 lncRNAs targeted 9874 genes, and within a 50–100 kb range, 1878 lncRNAs targeted 11,716 genes (Appendix A). Conducting GO analysis on the target genes of DELs within a 100 kb range in each comparison group, significant enrichment terms were observed. In the P vs. W group, enriched terms included ADP binding (GO:0043531), FAD binding (GO:0071949), and signal transduction (GO:0007165), among others (Appendix A). In the Y vs. P group, enriched terms comprised peroxisome (GO:0005777), metal ion transport (GO:0030001), and electron transport chain (GO:0022900) (Appendix A). Finally, the Y vs. W group showed enrichment in terms such as ADP binding, hexosyltransferase activity (GO:0016758), and response to auxin (GO:0009733), among others (Appendix A).

### 2.4. lncRNA-miRNA-mRNA Regulatory Network Associated with the Cytoskeleton Organization in Coma

The construction and arrangement of the cell cytoskeleton form the foundation of coma elongation, with microtubules and microfilaments comprising the cell cytoskeleton. Kinesin and actin, two crucial proteins in the cellular cytoskeleton system, play significant roles during the development of coma. In this study, significant expression of eight kinesin-encoding genes was observed during the initial stage of coma development. They were targeted by 27 lncRNAs, including 20 trans-acting lncRNAs and 7 cis-acting lncRNAs (Appendix A). Integration of degradome sequencing data and predictions from psRNAtarget identified an lncRNA-miRNA-mRNA regulatory network involving 6 kinesin-encoding transcripts, with 5 miRNAs and 20 lncRNAs (Figure 5A). Furthermore, four actin-encoding genes were found to exhibit significant upregulation during the initial stages of coma development. They were targeted by 14 lncRNAs, with 10 trans-acting lncRNAs and 4 cis-acting lncRNAs (Appendix A). Further investigation revealed that 5 actin-encoding transcripts, along with 7 miRNAs and 44 lncRNAs, constitute an lncRNA-miRNA-mRNA regulatory network (Figure 5B). These lncRNAs actively participate in coma development by indirectly regulating the cellular cytoskeleton system.

### 2.5. lncRNA-miRNA-mRNA Regulatory Network Associated with the Synthesis and Remodeling of the Cell Wall in Coma

In coma, the correct arrangement of the cytoskeleton facilitates the directed synthesis and accumulation of cell wall materials, serving as a prerequisite for coma elongation. Cellulose synthase and sucrose synthase play pivotal roles in the synthesis of cell wall materials. In this study, differential expression of 17 genes encoding cellulose synthase was found during coma development, with 15 differentially expressed cellulose synthase-encoding genes targeted by 45 lncRNAs, including 29 trans-acting lncRNAs and 16 cis-acting lncRNAs (Appendix A). Further investigation revealed that 8 cellulose synthase-encoding transcripts, with 8 miRNAs and 39 lncRNAs, constitute an lncRNA-miRNA-mRNA regulatory network (Figure 6A). Notably, Potri.003G142500, Potri.002G257900, and Potri.006G251900 exhibited the highest expression levels in the late stages of coma elongation, while Potri.018G103900 showed the highest expression levels in the early stages of coma elongation. Additionally, five genes encoding sucrose synthase exhibited differential expression during coma development, with four differentially expressed sucrose synthase-encoding genes targeted by 14 lncRNAs, including 12 trans-acting lncRNAs and 2 cis-acting lncRNAs (Appendix A). Further investigation revealed that 4 sucrose synthase-encoding transcripts, with 3 miRNAs and 11 lncRNAs, constitute an lncRNA-miRNA-mRNA regulatory network (Figure 6B). Notably, Potri.002G202300 exhibited the highest expression levels in the early stages of coma elongation, while Potri.004G081300 showed the highest expression levels in the late stages of coma elongation. The elongation of coma cells is also dependent on the plasticity of the cell wall, where expansin plays a crucial role in the process of cell wall remodeling. In this study, fifteen differentially expressed expansin-encoding genes were targeted by 51 lncRNAs, including 25 trans-acting lncRNAs and 15 cis-acting lncRNAs (Appendix A). Further investigation revealed that 3 expansin-encoding transcripts, with 4 miRNAs and 10 lncRNAs, constitute an lncRNA-miRNA-mRNA regulatory network (Figure 6C). Notably, Potri.010G202500, Potri.009G141400, Potri.016G135200, and Potri.003G083200 exhibited the highest expression levels in the late stages of coma elongation.

### 2.6. Analysis of the lncRNA-miRNA-TF Regulatory Relationships during Coma Development

Transcription factors, a type of protein molecule with a unique structure that regulates gene expression, play a crucial role in various stages of plant growth and development. The MYB, bHLH, and WDR transcription factor families are essential components of the initial regulatory model for development in Arabidopsis trichomes and cotton fibers. A similar regulatory mechanism is hypothesized in the initial stages of poplar coma development. During the W period, 54 bHLH members showed significantly elevated expression, with 49 targeted by 127 lncRNAs. Of these, 88 lncRNAs exhibited trans-acting, while 41 displayed cis-acting regulation (Appendix A). Integrating degradome sequencing data and predictions from psRNAtarget identified an lncRNA-miRNA-mRNA regulatory network involving 13 bHLH members, 77 lncRNAs, and 24 miRNAs (Figure 7). AtGL3 and AtEGL3, two pivotal members of the bHLH transcription factor family involved in the initiation regulation of trichomes in Arabidopsis, phylogenetic analysis revealed close relationships between *AtGL3* and *AtEGL3* in Arabidopsis and *Potri.003G128000.3* and *Potri.001G103600.2* in poplar (Appendix A). The regulatory network revealed that miR7817-5p targets *Potri.003G128000.3* while simultaneously targeting nine lncRNAs.

Furthermore, 48 members of the R2R3 MYB family exhibited significantly elevated expression during the W period, with 43 of these R2R3 MYB members being targeted by 99 lncRNAs. This includes 68 lncRNAs with a trans-acting and 31 lncRNAs with a cis-acting (Appendix A). Further investigation revealed that 5 members of the R2R3 MYB family, along with 11 lncRNAs and 7 miRNAs, formed an lncRNA-miRNA-mRNA regulatory network (Appendix A).

In addition, 56 members of the Transducin/WD40 repeat-like superfamily protein family showed high expression during the W period, targeted by 67 lncRNAs. Among these, 36 members were targeted by 67 lncRNAs, including 41 with a trans-acting and 26 with a cis-acting (Appendix A). Further investigation unveiled that 6 members of the Transducin/WD40 repeat-like superfamily protein family, along with 51 lncRNAs and 7 miRNAs, constituted an lncRNA-miRNA-mRNA regulatory network (Appendix A). AtTTG1, a core member of the MBW transcriptional activation complex, belongs to the Transducin/WD40 repeat-like superfamily protein family. In this study, close phylogenetic relationships were observed between *AtTTG1* and two members of the *P. trichocarpa* Transducin/WD40 repeat-like superfamily protein family, namely *Potri.012G006100.3* and *Potri.012G006100.4* (Appendix A). Intriguingly, neither of them exhibited significant upregulation during the initiation stage of poplar coma development. This suggests that they may not play a crucial role in the regulation of coma initiation.

## 3. Discussion

In recent years, lncRNAs have emerged as crucial regulators of gene expression in plant growth, development, and stress responses, attracting considerable attention for the exploration of their regulatory mechanisms and functions [41]. However, there has been a notable gap in research examining the role of lncRNAs in regulating poplar coma development. The objective is to identify genes that may influence the initiation and progression of poplar coma and to uncover potential regulatory associations between lncRNAs and these genes. Through high-throughput sequencing of three distinct developmental stages of poplar coma, a total of 774,841,556 clean reads were obtained from nine libraries, and a total of 1888 lncRNAs and 52,810 mRNAs were identified. This includes 474 antisense lncRNAs, 830 intergenic lncRNAs, 562 sense overlapping lncRNAs, and 22 intronic lncRNAs. Notably, intronic lncRNAs consistently exhibit minimal average length, exon number, and ORF length, while antisense lncRNAs consistently show the lowest expression levels throughout all coma development stages. Compared with mRNAs, lncRNAs are characterized by shorter lengths, lower expression levels, and fewer exons. These findings align with earlier reports on various species, including poplar, longan, and ginkgo [42,43,44]. Given the protracted generation cycles of trees, research on molecular mechanisms during reproductive growth stages lags, particularly in the context of poplar coma development. Nevertheless, comprehensive studies on the regulatory mechanisms governing trichome initiation and development in Arabidopsis leaf and cotton seed offer valuable insights as a reference for this research. A model has been developed to describe the regulatory mechanisms at different stages of coma development (Figure 8).

### 3.1. lncRNAs Interacting with TFs during the Initiation Stage of Poplar Coma

The C2H2 ZFP family plays the most upstream role in the cell fate determination model. As a member of the C2H2 ZFP family, AtGIS functions downstream of the GA signaling pathway, controlling epidermal differentiation by modulating the activity of transcription factors implicated in trichome initiation. *AtGIS* overexpression leads to a substantial increase in trichome density on floral organs, stems, and branches, while *AtGIS* mutants exhibit the opposite phenotype [45]. Additional family members, including AtZFP5, AtZFP6, AtZFP8, and AtGIS2, also play pivotal roles in the initiation process of the Arabidopsis trichome [46,47]. AtGL3 and AtEGL3 are bHLH family members initially identified in Arabidopsis [48]. They form a transcription activation complex by interacting with WDR and R2R3-MYB transcription factors, thereby regulating the initiation of trichome development in Arabidopsis. *AtGL3* and *AtEGL3* demonstrate functional redundancy, with *AtGL3* mutations significantly impacting trichome development, resulting in reduced nuclear replication, decreased branching, and smaller trichome, while exerting a minor effect on trichome quantity [49]. Plants with double mutations in *AtGL3* and *AtEGL3* exhibit a complete absence of trichomes [50]. Based on this research, among the 242 bHLH family members, *Potri.003G128000.3* and *Potri.001G103600.2* were identified as having the closest phylogenetic relationship with AtGL3 and AtEGL3. Both genes are significantly upregulated during the initiation stage of coma. TCONS_00016788, TCONS_00033583, and TCONS_00098833 target *Potri.003G128000* through trans-acting mechanisms, while TCONS_00008661, located 236 bp upstream, targets *Potri.001G103600* through cis-acting mechanisms. Additionally, the targeting of Potri.003G128000.3 by miR7817-5p was also observed, along with the simultaneous targeting of nine lncRNAs. The aforementioned lncRNAs may interact with specific bHLH transcription factors through different mechanisms, thereby participating in the regulation of the initiation of coma. In the R2R3 MYB family, the AtGL1 member of the SGB15 subfamily is crucial for trichome initiation in Arabidopsis, with mutations resulting in a non-trichome phenotype in leaves [51]. Previous studies have indicated the absence of SGB15 subfamily members in the R2R3 MYB family of poplar [3,52]. As one of the members of the MBW transcription activation complex, mutations in *AtTTG1* result in a non-trichome phenotype in Arabidopsis leaves [53]. Identified in the poplar Transducin/WD40 repeat-like superfamily proteins were *Potri.012G006100.3* and *Potri.012G006100.4*, which have the closest phylogenetic relationship with *AtTTG1* among 243 members. However, *Potri.012G006100* did not exhibit significant differential expression at various stages of coma development, suggesting this gene may not be involved in the regulation of coma initiation.

### 3.2. lncRNAs Involvement in Constructing and Arranging Coma Cytoskeleton

Much like the developmental processes observed in Arabidopsis trichomes and cotton fibers, poplar coma development entails significant cell expansion, where the plant cell cytoskeleton emerges as a pivotal determinant guiding the direction of this expansion [54,55]. It is noteworthy that the GO analysis of the target genes affected by DELs during poplar coma development revealed significant enrichment of terms such as kinesin complex, microtubule-based movement, and microtubule motor activity. Plant motor proteins harness energy from ATP hydrolysis to drive movement along microtubules and microfilaments, not only facilitating the directed transport of substances like proteins and organelles within the cell but also orchestrating the arrangement and distribution of microtubules and microfilaments in plant cells [56]. This regulatory capacity contributes to the formation and maintenance of cell polarity structures, governing the direction of deposition of newly synthesized cell wall materials and playing a pivotal role in cell expansion [57]. In Arabidopsis trichomes, there is a transversely arranged microtubule array at the apexes of elongating branches, with F-actin caps aligning parallel to the growth axis of the trichome [58]. Interestingly, an analogous cellular cytoskeleton arrangement was observed at the tip of elongating cotton fiber cells [54]. The Kinesin-like Calmodulin Binding Protein (KCBP), acting as a central hub integrating microtubules (MT) and actin filaments, exhibits a gradient distribution along cortical microtubules [58]. Mutations in *AtKCBP* disrupt the arrangement and distribution patterns of the F-actin cap and microtubule arrays, resulting in a significant reduction in the length of the trichome stem and a decrease in the number of branches [7]. In this study, a notable upregulation of two genes encoding KCBP, namely *Potri.001G455300* and *Potri.011G146700*, was observed during the W period. Three lncRNAs (TCONS_00033583, TCONS_00116874, and TCONS_00119782) were identified as trans-acting regulators targeting *Potri.001G455300*. Additionally, TCONS_00006780, positioned downstream by 30,287 kb from *Potri.001G455300*, exhibited cis-acting regulation on this gene. It was found that miR156a targets Potri.001G455300, but no simultaneous targeting of any lncRNA by this miRNA was identified. TCONS_00015959, TCONS_00069606, and TCONS_00000022 were found to act as trans-acting regulators, targeting *Potri.011G146700*. Moreover, silencing *GhACT1* through RNA interference disrupts the actin cytoskeleton in cotton fibers, leading to the inhibition of fiber elongation [59]. Based on these findings, DEGs encoding kinesin and actin were found to be highly expressed during the initial stage of coma, suggesting that the construction and arrangement of the cell cytoskeleton during coma elongation are likely predominantly carried out by proteins synthesized in the initial stages. Additionally, the lncRNAs targeting genes encoding kinesin and actin through different mechanisms were identified, indicating their involvement in poplar coma development by indirectly regulating the construction and arrangement of the cell cytoskeleton.

### 3.3. lncRNA Is Involved in the Synthesis and Remodeling of Coma Cell Walls

During the process of cell wall expansion and cell elongation, expansin (EXP) can reshape the internal structure of the cell wall, leading to softening and relaxation. Ultimately, turgor pressure drives cell elongation [60]. GbEXPATR is a unique expansin found in *Gossypium barbadense*. Overexpression of this gene stimulates the elongation of cotton fibers, leading to cells with thinner cell walls and increased fiber strength [18]. Similarly, the overexpression of *GhEXPA8* and *GhEXPA1* also contributes to an increase in the length of cotton fibers [61,62]. In this study, a significant upregulation of *Potri.016G135200* (EXP A8) was observed during the Y period. Three lncRNAs (TCONS_00107723, TCONS_00105633, and TCONS_00105631) were identified as targeting this gene through cis-acting mechanisms. The simultaneous targeting of this gene and five lncRNAs by miRN30 was also found. In the elongation process of fiber cells, the formation and remodeling of the primary cell wall, as well as the subsequent thickening of the secondary cell wall after elongation cessation, involve the deposition of a significant amount of wall materials [10,15,63]. Plant cell walls consist primarily of three polysaccharides: cellulose, hemicellulose, and pectin [64]. Cellulose synthases use UDP-glucose as a substrate to add glucose units to the cellulose chain, and the substrate UDP-glucose can be synthesized through the action of enzymes such as sucrose synthase or UDP-Glc pyrophosphorylase. Knockouts of *GhCesA4*, *GhCesA7*, and *GhCesA8* impede the formation of secondary cell walls in cotton fibers, resulting in shortened fiber length and loss of the ability to twist [65]. In this investigation, significant upregulation of *Potri.003G142500* (cellulose synthase-like G2) was observed during the Y period, and significant upregulation of *Potri.018G103900* (Cellulose synthase family protein) was observed during the P period. TCONS_00113860, TCONS_00052482, TCONS_00063173, and TCONS_00024892 were identified as trans-acting regulators targeting *Potri.003G142500*, while TCONS_00024992 targeted *Potri.003G142500* through cis-acting mechanisms. miR476 simultaneously targets this gene and 10 lncRNAs. TCONS_00113860, TCONS_00116215, and TCONS_00113865 target *Potri.018G103900* through cis-regulation. miRN14 simultaneously targets this gene and 11 lncRNAs, while miR167a simultaneously targets this gene and 3 lncRNAs. Furthermore, the inhibition of sucrose synthase expression suppresses the initiation and elongation of cotton fiber cells [66]. Conversely, overexpression of sucrose synthase in cotton results in fiber elongation and thickening of secondary cell walls [67]. In this study, high expression of *Potri.002G202300* (Sucrose Synthase 3) was observed during the P period, and high expression of *Potri.004G081300* (sucrose synthase 5) was observed during the Y period. TCONS_00106139, TCONS_00015901, and TCONS_00111795 target *Potri.002G202300* through trans-regulation. TCONS_00084439, TCONS_00050202, and TCONS_00009217 target *Potri.004G081300* through trans-regulation, while TCONS_00030789 targets this gene through cis-regulation. The aforementioned lncRNAs act on the target genes in different ways, participating in the elongation of poplar coma by indirectly regulating the synthesis and remodeling of the cell wall.

## 4. Materials and Methods

### 4.1. Plant Materials and Sampling

The collected mature branches of ‘nanlin895′ (*Populus deltoides* × *Populus euramericana*) were placed in room-temperature water for cultivation. During the hydroponic process, florets were collected from the branches, and the development of internal coma was observed under a microscope, categorizing them into three periods. The collected florets from three different periods were immediately placed into liquid nitrogen and subsequently stored at −80 °C for future use. In the W period, the epidermal cells of the placenta and funicle were indistinguishable from the surrounding cells, and the coma had not yet formed. A complete ovary structure could be observed, with the funicle diameter slightly smaller than the ovule at this stage. In the P period, significant outward protrusions of the epidermal cells of the placenta and funicle formed the coma, and at this point, the funicle diameter was equivalent to that of the ovule. In the Y period, a substantial and elongated coma was observed, tightly enveloping the ovule and filling the entire ovary. At this stage, the funicle diameter was greater than that of the ovule [3].

### 4.2. Construction and High-Throughput Sequencing of Strand-Specific Libraries

The RNAPrep Pure Plant Plus Kit (TIANGEN^®^, Beijing, China) was utilized for extracting total RNA from all plant materials. Subsequently, RNA degradation or contamination was assessed through electrophoresis on a 1.2% agarose gel. The purity of the RNA was determined using the Nanophotometer^®^ spectrophotometer (Implen, Westlake Village, CA, USA), while the Agilent Bioanalyzer 2100 system (Agilent Technologies, Santa Clara, CA, USA) was employed to evaluate the integrity of the RNA. A method for constructing strand-specific libraries was employed, which involved the removal of ribosomal RNA (rRNA) [68]. Initially, rRNA was depleted from total RNA, and the fragmented RNA served as a template. Random oligos were used as primers to synthesize the first strand of cDNA, with dUTP replacing dTTP during the synthesis of the second strand of cDNA. Following these steps, the double-stranded cDNA underwent end repair, A-tailing, and the ligation of sequencing adapters. AMPure XP beads (Beckman Coulter, Brea, CA, USA) were utilized to selectively isolate cDNA fragments of approximately 350~400 bp. The second strand of cDNA, which contained uracil (U), was enzymatically degraded using the USER enzyme, and the resulting library was generated through PCR amplification. Subsequent to the successful validation of the library, high-throughput sequencing of the sample libraries was conducted using the Illumina HiSeq platform.

### 4.3. Read Mapping and Transcriptome Assembling

Firstly, the raw data was quality controlled using FASTP software (v.0.23.4), which removed sequencing adapters and filtered out low-quality reads. Utilizing the HISAT2 (v2.2.0) software, clean reads were aligned to the reference *Populus trichocarpa* genome (v4.1) [69]. The SAMTOOLS (v1.20) parameter ‘F 256’ was used to filter out reads with multiple alignments. Subsequently, the STRINGTIE (v2.2.1) software was employed to assemble reads into transcripts [70], and the CUFFLINKS (v2.2.1) software was used to merge transcripts assembled from individual samples. Finally, quantitative analysis of transcripts expression levels was conducted using Fragments Per Kilobase of exon model per Million mapped fragments (FPKM) [71]. Differential expression analysis was performed using the R package DESEQ2, with criteria set as padj < 0.05 and |log2 (fold change)| > 1 to identify statistically significant differences.

### 4.4. Identification of lncRNA and Prediction of Its Target Genes

Filtering out transcripts with a length not exceeding 200 nucleotides, Coding Potential Calculator (CPC2) (http://cpc.cbi.pku.edu.cn/ (accessed on 13 September 2023)) [72], Protein Family Database (PFAM) (http://pfam.xfam.org/ (accessed on 13 September 2023)) [73], Coding–Non-Coding Index (CNCI) (http://www.bioinfo.org/software/cnci (accessed on 13 September 2023)) [74] and A Tool for Predicting Long Non-coding RNAs and Messenger RNAs Based on an Improved K-Mer Scheme (PLEK) (http://sourceforge.net/projects/plek/ (accessed on 13 September 2023)) [75] were employed for coding potential prediction. The intersection of transcripts predicted to have coding potential by these tools was taken as the candidate lncRNAs. Following the naming guidelines provided by the HUGO Gene Nomenclature Committee (HGNC) (https://www.genenames.org/ (accessed on 21 September 2023)) for lncRNA, systematic naming of lncRNAs was proceeded. Following this, an expression level filtering step was implemented on the previously predicted candidate lncRNAs. Transcripts with FPKM values of ≥0.5 for those with multiple exons and FPKM values of ≥2 for single-exon transcripts were specifically chosen. This process led to the final set of identified lncRNAs.

Two approaches were utilized to predict the target genes of lncRNAs. In the first method, co-location analysis was employed to predict target genes based on the positional relationship between lncRNA and protein-coding genes. Candidate target genes for cis-regulation were selected from protein-coding genes within a 100 kb range upstream and downstream of the lncRNA [42,76,77]. The second method involved co-expression analysis, predicting target genes based on the expression correlation between lncRNA and protein-coding genes. Using an absolute correlation coefficient greater than 0.97 as the screening criterion, the top three lncRNAs were identified as candidate target genes for trans-regulation. For Gene Ontology (GO) analysis, the R package “CLUSTERPROFILER” was employed. All genes annotated in the GO were considered background genes, and the target genes of differentially expressed lncRNAs (DELs) were designated as foreground genes for hypergeometric distribution testing. GO terms with a *p*-value ≤ 0.05 were considered significantly enriched.

### 4.5. Network Visualization

To construct the lncRNA-miRNA-mRNA regulatory network, the utilization of sRNA sequencing and degradome sequencing data was employed to reveal the targeting relationships between miRNA and mRNA. Afterward, lncRNA and miRNA sequences were submitted to the psRNATarget online platform (http://plantgrn.noble.org/psRNATarget/ (accessed on 27 September 2023)) [78], and prediction results with an expectation value ≤ 4 were used. Finally, mRNA and lncRNA targeted by the same miRNA are identified. The relationship diagrams among miRNA, lncRNA, and mRNA were drawn using CYTOSCAPE (v3.10.0). The relationship diagram between lncRNA and protein-coding genes was created using GEPHI (v0.10.0).

Information on the bHLH family members of *P. trichocarpa* was obtained from previously published articles [79]. The annotation files for *P. trichocarpa* (v4.1), where members of the Transducin/WD40 repeat-like superfamily protein family were searched, were retrieved from the PHYTOZOME website (https://phytozome-next.jgi.doe.gov/ (accessed on 1 October 2023)). The protein sequences of relevant transcription factors in *Arabidopsis thaliana* and cotton were downloaded from the National Center for Biotechnology Information (NCBI) website (https://www.ncbi.nlm.nih.gov/ (accessed on 2 October 2023)). Amino acid sequence alignments were performed using the MAFFT (v7.520) software, and the phylogenetic tree was constructed using the approximately maximum-likelihood method implemented in FASTTREE (v2.1.11). The resulting phylogenetic tree was beautified using tools provided on the Interactive Tree of Life (iTOL) website (https://itol.embl.de/ (accessed on 5 October 2023)).

### 4.6. Real-Time Quantitative PCR Validation

All RT-qPCR experiments were conducted on the ABI ViiA 7 Real-Time PCR System (Thermo Fisher, Waltham, MA, USA). SYBR Premix Ex Taq (TaKaRa, Tokyo, Japan) was used for RT-qPCR detection following the manufacturer’s instructions. Each reaction included three technical replicates, and *PeEF1α* was used as the housekeeping gene [80]. The relative expression levels of genes were calculated using the 2^−ΔΔCt^ method [81]. Primers for all lncRNAs were designed using Beacon Designer (v7) software, and the specific primer sequences can be found in Appendix A.

## 5. Conclusions

This study constructed a chain-specific library and conducted high-throughput sequencing to identify lncRNAs and mRNAs for the first time during the development of poplar coma. Analyzing the dynamic changes in the transcriptome during coma development, the expression profiles of lncRNAs and mRNAs were investigated. Additionally, a regulatory network involving lncRNA-miRNA-TF in coma initiation was established. Furthermore, we analyzed the lncRNA-miRNA-mRNA networks regulating cell cytoskeleton organization and the synthesis/remodeling of cell walls during coma elongation. These findings offer valuable insights for a more profound comprehension of the molecular mechanisms governing poplar coma development.

## Figures and Tables

**Figure 1 ijms-25-07403-f001:**
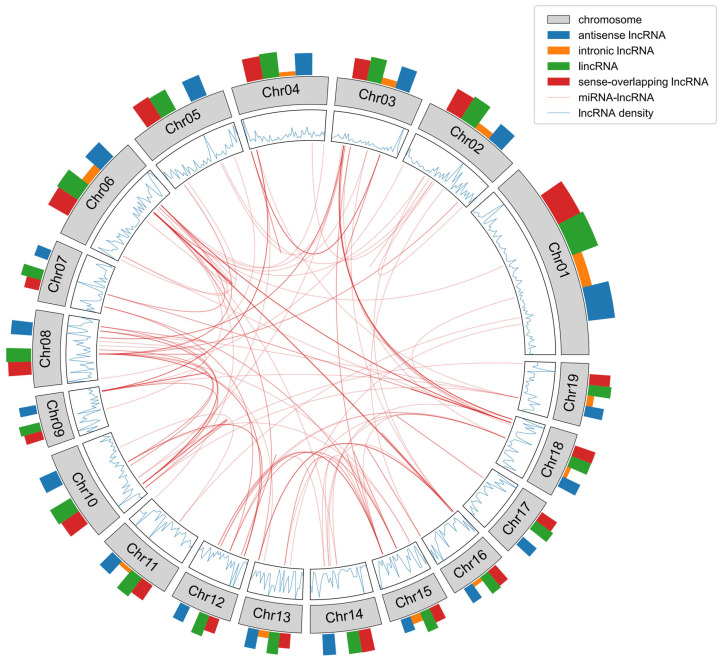
Distribution of lncRNAs on chromosomes. The outermost bar chart represents the quantity of different types of lncRNA on each chromosome; the blue curve represents the density of lncRNA in different regions of each chromosome; the red curve indicates miRNA-lncRNA regulatory relationships predicted by the psRNAtarget online website (with an expectation < 3).

**Figure 2 ijms-25-07403-f002:**
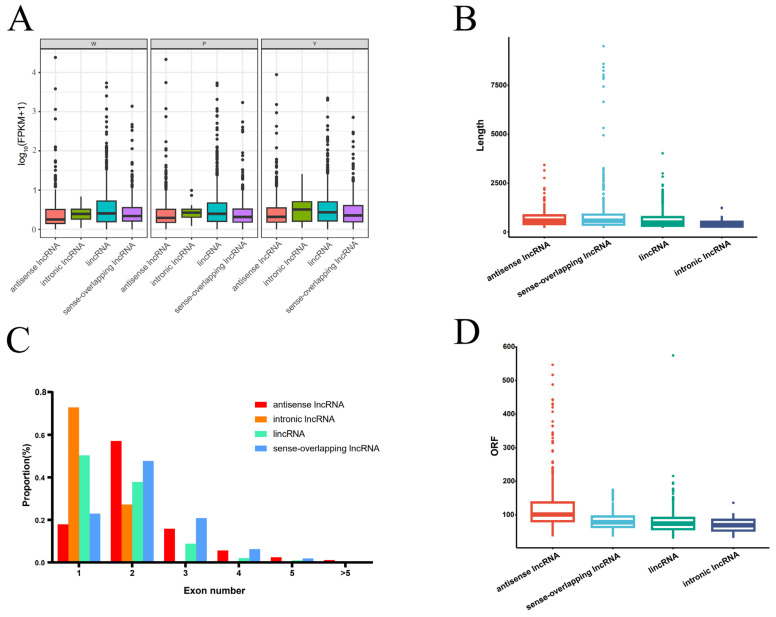
Comparative features of different types of lncRNA. (**A**). Comparison of the expression levels of different types of lncRNAs in three developmental stages of poplar coma. (**B**). Comparison of the lengths between different types of lncRNA. (**C**). Comparison of the number of exons between different types of lncRNA. (**D**). Comparison of ORF lengths among different types of lncRNA.

**Figure 3 ijms-25-07403-f003:**
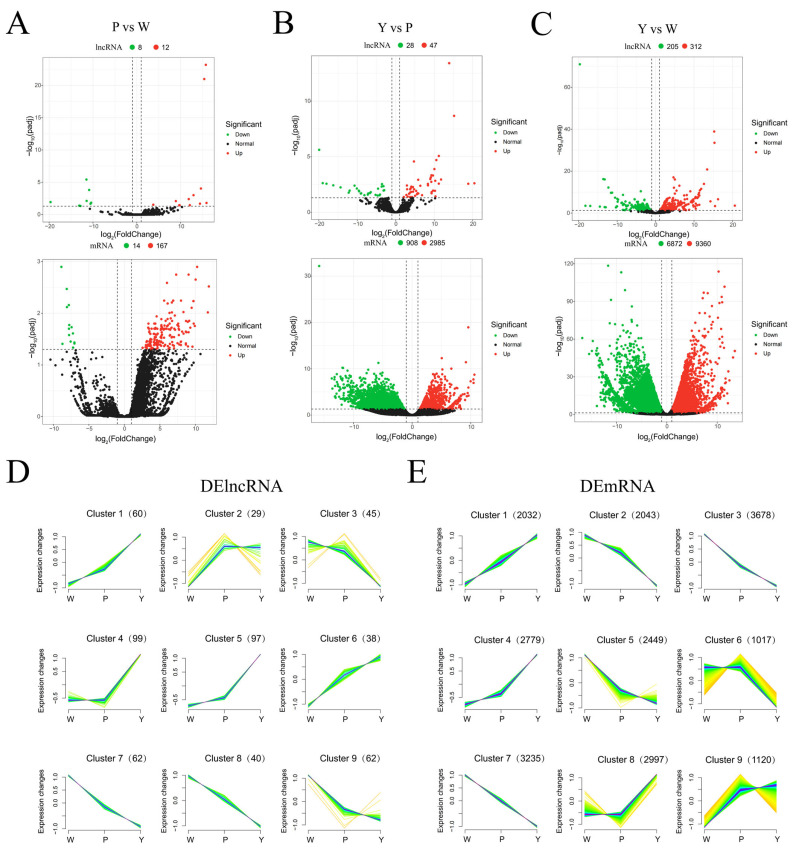
Expression patterns of lncRNAs during different developmental stages of poplar coma. (**A**). Volcano plot of DELs and DEGs in the PvsW comparison group. (**B**). Volcano plot of DELs and DEGs in the YvsP comparison group. (**C**). Volcano plot of DELs and DEGs in the YvsW comparison group. The upper diagram represents lncRNA, while the lower diagram represents mRNA, red dots represent significantly upregulated genes, and green dots represent significantly downregulated genes. (**D**). Time series analysis of DELs. (**E**). Time series analysis of DEGs.

**Figure 4 ijms-25-07403-f004:**
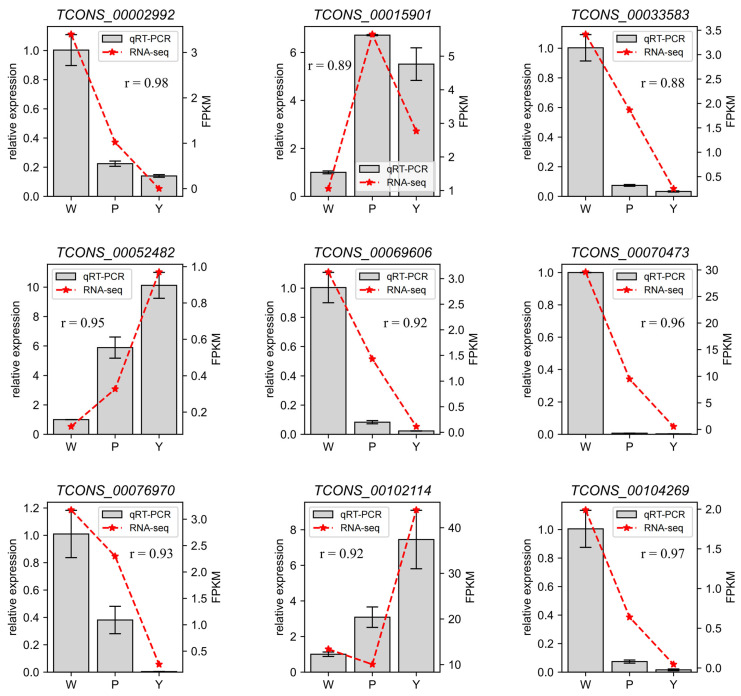
Sequencing data expression identified by RT-qPCR. The broken line shows the relative expression levels obtained by RT-qPCR (left *y*-axis). The bar graph shows reads FPKM values obtained by sRNA-seq (right *y*-axis). For the results obtained using sRNA-seq and RT-qPCR, the expression levels at each developmental stage were normalized to the mean expression levels across the three biological replicates.

**Figure 5 ijms-25-07403-f005:**
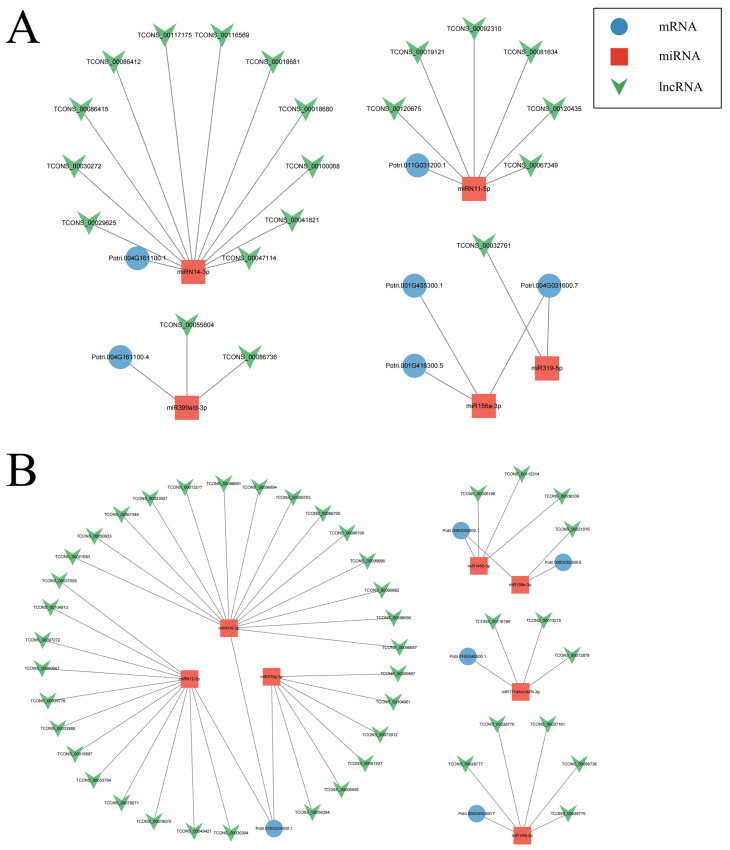
The lncRNA-miRNA-mRNA regulatory network associated with the cytoskeleton organization of coma. (**A**). The lncRNA-miRNA-mRNA regulatory network constituted by kinesin-encoded transcripts. (**B**). The lncRNA-miRNA-mRNA regulatory network constituted by actin-encoded transcripts. The blue circle represents mRNA, the red square represents miRNA, and the green triangle represents lncRNA.

**Figure 6 ijms-25-07403-f006:**
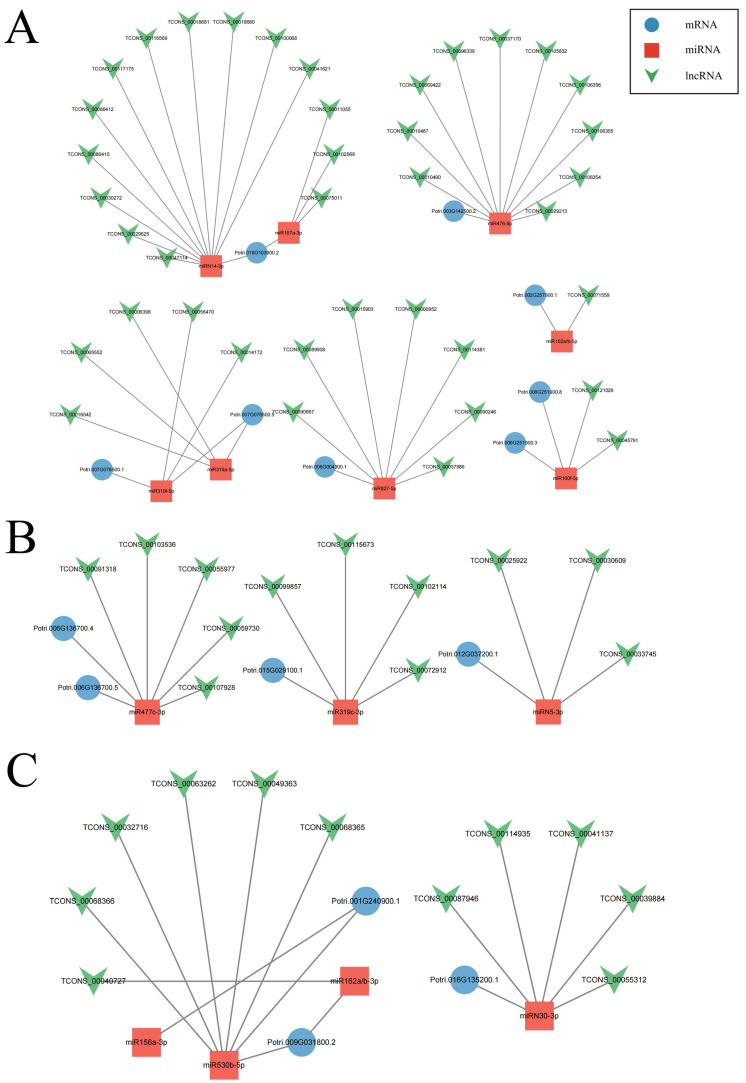
The lncRNA-miRNA-mRNA regulatory network associated with cell wall synthesis and remodeling in coma. (**A**). The lncRNA-miRNA-mRNA regulatory network is constituted by cellulase synthase-encoded transcripts. (**B**). The lncRNA-miRNA-mRNA regulatory network is constituted by sucrose synthase-encoded transcripts. (**C**). The lncRNA-miRNA-mRNA regulatory network is constituted by expansin-encoded transcripts.

**Figure 7 ijms-25-07403-f007:**
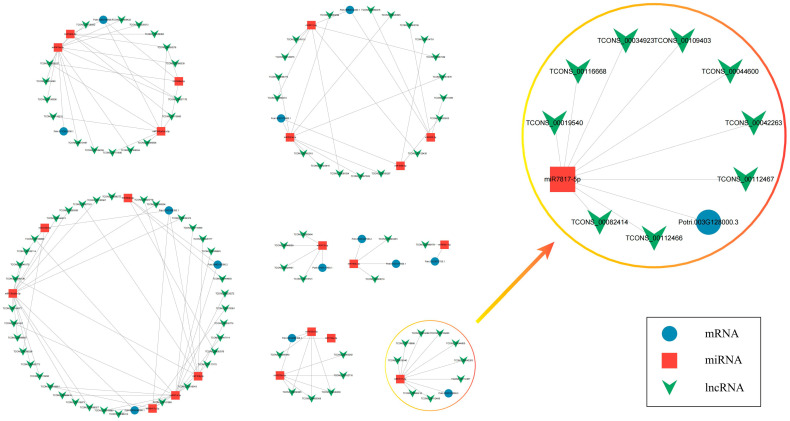
lncRNA-miRNA-bHLH regulatory networks. The blue circle represents mRNA, the red square represents miRNA, and the green triangle represents lncRNA.

**Figure 8 ijms-25-07403-f008:**
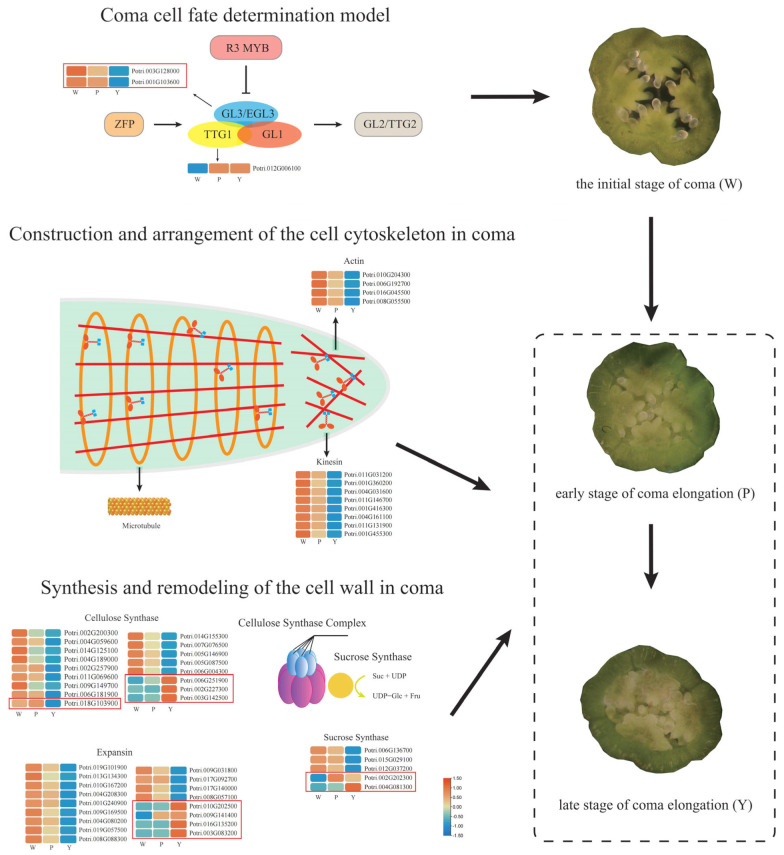
Regulatory mechanism model at different developmental stages of poplar coma.

## Data Availability

The data related to the article has been submitted to NCBI, with the project number PRJNA1129894.

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
