# Peer review of "Unraveling the lncRNA-miRNA-mRNA Regulatory Network Involved in Poplar Coma Development through High-Throughput Sequencing"

_ijms, 2024, doi:10.3390/ijms25137403_

Round 1
Reviewer 1 Report
Comments and Suggestions for Authors
Unraveling the lncRNA-miRNA-mRNA Regulatory Network Involved in Poplar Coma Development Through High-throughput Sequencing
In this manuscript, Song et al. investigate the molecular mechanisms underlying the development of poplar coma. Understanding these fundamental mechanisms is crucial for various reasons, including, as the authors mention, alleviating potential environmental pollution and health risks. The authors aim to achieve this goal by employing strand-specific RNA-seq to identify lncRNAs and mRNAs during poplar coma development. They integrate multi-layer RNA data to create lncRNA-miRNA-mRNA regulatory networks during this process. Song et al performed the lncRNA analysis which I would consider the main novel factor for this paper compared to their previous work including Xu et al 2023: doi: 10.1016/j.isci.2023.106496. I acknowledge that while lncRNAs were once considered mere noise, they have gained recognition as important regulators of biological functions and important molecular players that need to be studied further. Although there are no fundamental issues with the paper, several areas need to be addressed and improved before publication.
· Which version of Populus trichocarpa genome were used? Additionally, is this genome relevant for the model used, i.e., ‘Nanlin895’ (Populus deltoides × Populus euramericana)? If the genome for this specific hybrid is not available, it might be more appropriate to use one of its parental lines, as the Populus deltoides genome is publicly available. The authors should either provide a solid rationale for using Populus trichocarpa or reanalyze the data using a more relevant genome.
· Figure 8: Please replace the Accession/Gene ID numbers with the corresponding gene names, or at least provide the gene names in parentheses. Additionally, please include a legend in the heatmap to correlate the colors with the expression levels. In general, all figures should be of higher resolution. This might be an issue related to the review copy, but if not, there are lots of instances where the text in the figures is not properly legible and needs improvement.
· In line 524, the phrase "After conducting quality control on the raw data…" needs further elaboration to explain the specific steps taken for quality control.
· For the alignment, please specify the parameters used. How do you handle potential issues with reads that align to multiple locations or reads that do not align well to the reference genome? All tools utilized should be described in sufficient detail to enable others to replicate the results.
· It appears that DE results were combined from BALLGOWN and DESEQ2. Please explain why both tools were used and how their results were integrated or compared.
· Lines 50-51: "To address this issue fundamentally, it is crucial to investigate the molecular mechanisms underlying the development of poplar coma." It seems the authors imply that understanding the mechanism of coma development could help mitigate its negative impacts, perhaps by making molecular-level alterations. However, it's important to consider whether altering the coma would affect the development of the capsule or the discharge of seeds. If there is a significant impact, the authors should reframe the central objective to reflect the broader implications and benefits of their research.
· Line 21 ‘Analyzed were the expression profiles of lncRNAs and mRNAs during coma development.’ I don’t think the sentence structure is correct.

Comments on the Quality of English LanguageAuthor Response
Please see the attachment.

Reviewer 2 Report
Comments and Suggestions for Authors
The ms by Song et al. reports transcriptome profiles from three stages of the coma development in poplar (Populus deltoides x P. euramericana). Using gene differential expression in mRNA, miRNA and long noncoding RNA, the authors attempt to infer gene networks at the basis of the cell wall synthesis and other aspects of the coma development.
The results from this ms are certainly interesting and worth publishing. My major concern is the weak inference of gene networks, that is mainly based on expression correlation and location of interacting RNAs. No inference of causal relationships is attempted in this study.
For cis-regulation, the inference of gene regulation was based on proximity of protein-coding genes (within 100Kb upstream and downstream of lncRNA). Was this the only criterion? At the very least, both proximity and correlation in expression level should be used for such inference.
For trans-regulation, co-expression analysis was used. For the interaction with microRNA, the authors mentioned the use of degradome sequencing data and sRNA sequencing, but this needs to be further explained.
Given the weak approaches for gene network inference and the lack of evidence for causal relationships, the results on gene networks between lncRNA, miRNA and mRNA need to be toned down, starting from the title. It should be specified that the figures 5, 6 and 7 are not causal gene networks, and that they are co-expression networks, based on similar expression patterns/genomic co-location.
Minor comments:
Fig 2: Font for X axis labels are too small.
Fig 3. Font for figure legends and axes labels are too small
Comments on the Quality of English Language
I have found few places where the English can be improved. For example:
Line 124. “In poplar, samples..” change to “in poplar, RNA samples..”
Line 495. “Three periods”, change to “three stages”
Round 2
Reviewer 2 Report
Comments and Suggestions for Authors
The authors have provided some justified, but also non-justified, responses to my criticisms.
The prediction of cis-regulatory lncRNA was based on proximity of target genes. The authors state that this is an adequate standard procedure and provide a list of references. At the very least, the authors should make the changes to the methodology part and cite the relevant literature that was listed in the responses.
However, the use of this approach does provide a high rate of false positives (see for example Dhaka et al. Nucleic Acid Research 52, 2821-2835, 2024). The authors might want to consider the statistical methods described in Dhaka et al. for improving prediction of cis-regulatory lncRNA.
Author Response
Comments 1: The prediction of cis-regulatory lncRNA was based on proximity of target genes. The authors state that this is an adequate standard procedure and provide a list of references. At the very least, the authors should make the changes to the methodology part and cite the relevant literature that was listed in the responses.
Response 1: Thank you for pointing this out. I have already cited the relevant references in Section 4.4 of the Materials and Methods section of the article. The modified content is as follows:
Two approaches were utilized to predict target genes of lncRNAs. In the first method, co-location analysis was employed to predict target genes based on the positional relationship between lncRNA and protein-coding genes. Candidate target genes for cis-regulation were selected from protein-coding genes within a 100kb range upstream and downstream of the lncRNA [42, 76-77].
Comments 2: However, the use of this approach does provide a high rate of false positives (see for example Dhaka et al. Nucleic Acid Research 52, 2821-2835, 2024). The authors might want to consider the statistical methods described in Dhaka et al. for improving prediction of cis-regulatory lncRNA.
Response 2: Thank you very much for your valuable feedback. I have carefully read the article by Dhaka et al., and here are my thoughts: The software is currently applicable to humans and mice, but not to plants. Additionally, the regulatory mechanisms of lncRNAs are quite complex and vary significantly between different species, which have not been fully elucidated yet. Finally, this paper merely explores the potential target genes of lncRNAs, mainly focusing on the identification of lncRNAs and the construction of the lncRNA-miRNA-mRNA regulatory network during the development of poplar coma. This network construction is based on the targeting relationships between miRNA-lncRNA and miRNA-mRNA, with results obtained through degradome sequencing and online prediction tools, making the findings relatively accurate and reliable.